# Modeling Divisive Normalization as Learned Local Competition in Visual Cortex

## Abstract

Convolutional Neural Networks (CNNs) embody priors about the visual world: locality, stationary statistics, translation invariance, and compositionality. Similarly, CNNs implement the retinotopy of visual cortex—nearby pixels are processed by nearby neurons. A common cortical computation not usually included in CNNs is *divisive normalization*. It has been shown that divisive normalization of Gabor filters results in more statistically independent responses (Simoncelli & Heeger, 1998). In this paper, we model divisive normalization as a simple computationally-efficient layer that can be inserted at any stage within deep artificial neural networks. Divisive normalization acts on neuronal sub-populations, whose parameters are initialized from a multivariate Gaussian distribution. This leads to the emergence of learned competition between both orientation-preferring and color-opponent cell types. Divisive normalization improves categorization performance, as well as robustness to perturbed images. Interestingly, in smaller networks, divisive normalization as a non-linear operation eliminates the need for a non-linear activation function like ReLU to drive performance.

## 1 Introduction

Deep Convolutional Neural Networks (CNNs) have had much success as categorization models of real world visual stimuli, in particular on the ImageNet dataset (Deng et al., 2009; Krizhevsky et al., 2012; Simonyan & Zisserman, 2015; Szegedy et al., 2015; He et al., 2016). Moreover, they have been shown to be good at predicting neural responses to images in non-human and human ventral visual cortex (Yamins et al., 2014; Nayebi et al., 2018; Kubilius et al., 2019; Schrimpf et al., 2020). The superiority of CNNs over fully-connected networks can be attributed to their biologically-inspired properties of shared filter weights, translation invariance, and larger receptive fields with depth. These built-in priors reflect properties of the visual world: locality, stationary statistics, and the invariance of object identity with respect to translation.

Today, almost every CNN architecture makes use of some form of rectification, often ReLU, following its introduction by Glorot et al. (2011). While ReLU is not as biologically plausible as saturating nonlinearities, empirically, it works better in deep networks. ReLU models, however, do not account for one important nonlinearity—the common cortical computation of cross-feature inhibition (Simoncelli & Heeger, 1998). Carandini and Heeger (Carandini & Heeger, 2012) suggest that divisive normalization (DN) is a good model of this neural computation, and show that it can explain nonlinear properties of neurons in the primary visual cortex. Their model computes a ratio between the activity of an individual neuron and the summed activity of a local neighborhood of neurons. The activity of an individual neuron is produced after its summed input is halfwave-rectified and squared (Heeger, 1992; Wainwright et al., 2002). The reasons for performing DN include, but are not limited to, removing statistical dependence between pairs of filter responses obtained by projecting natural images onto linear basis functions at nearby spatial positions, orientations, and scales (Wainwright et al., 2002), light adaptation in the retina and contrast normalization (Boynton & Whitten, 1970), attention in the primary visual cortex (Reynolds & Heeger, 2009), and surround suppression through lateral excitation and inhibition (Cavanaugh et al., 2002; Carandini & Heeger, 2012).

As described above, divisive normalization in the VVC is often implemented in the form of local competition between sub-populations of neurons. This competition has been extensively observed between Gabor-like orientation selective cell types (Beck et al., 2011; Bonds, 1989; Busse et al.,

2009; Carandini et al., 1997; DeAngelis et al., 1992; Morrone et al., 1982). Recently, Cirincione et al. (2022) and Miller et al. (2022) implement divisive normalization in CNNs to demonstrate robustness to image corruptions and activation sparsity. However, they limit their analyses to either a single architecture, to competition only across V1 neurons, or to the introduction of an expensive divisive normalization computation.

In this paper, we propose a simple divisive normalization formulation inspired by Carandini & Heeger (2012) and a parameter initialization scheme based on the circular multivariate Gaussian distribution to drive local competition (section 3). This formulation of divisive normalization requires less computation than Miller et al. (2022), while still making the model robust to image corruptions to a large extent (sections 4.2 and 4.4). For shallower networks, we find that the use of divisive normalization as a squaring operator to turn the responses of neurons positive (making both excitatory and inhibitory signals excitatory) seems to be sufficient, as well as better, in terms of performance gains over using a half-wave rectifier before (section 4.1). We use various model architectures to robustly conclude the effectiveness of our divisive normalization formulation. And finally, we not only observe learned competition across Gabor-like orientation selective filters (as most works have previously shown), but also have this competition emerge between different *color-opponency* cell types (section 4.3).

## 2 RELATED WORK

There are three different normalization techniques worth thinking about that enforce some form of "competition" among neurons in some local neighborhood. The first is Local Response Normalization (Krizhevsky et al., 2012) that was proposed by the authors of AlexNet. This formulation uses a linear numerator and norms the responses of neurons by neighbors in some neighborhood over the channel space. Miller et al. (2022) then come up with a new formulation that is similar to LRN, but instead with a squared numerator and exponentially decaying weights attributed to neighbors away from the center in a neighborhood. They claim that their formulation is better than LRN in performance. Both of these normalization schemes use overlapping neighborhoods. Group normalization (Wu & He, 2018) is similar to our formulation of divisive normalization in the sense that it breaks down channels into mutually-exclusive (non-overlapping) neighborhoods, but then the computation is simply z-scoring the activations, rather than putting them in competition. Our approach differs from group normalization in the sense that it enforces local competition among neurons, as opposed to standardizing their responses.

There have been many prior works that have incorporated divisive normalization in deep networks: Jarrett et al. (2009) show that using non-linearities that include rectification and local contrast normalization is the single most important ingredient for good accuracy on object recognition benchmarks, but they only consider two layers. Ren et al. (2016) modified batch normalization and layer normalization by adding the additive constant in the denominator of the calculation like divisive normalization and explore how their version, which is also a canonical normalization, improves the performance of recurrent and convolutional neural networks for image classification. The numerator of their normalization formula is still first order, so it cannot remove the second order dependency as our formulation of divisive normalization can, whose numerator is squared (see equation 2). Pan et al. (2023) observes that CNN neurons are most suppressed when the surround matches the center and less suppressed when the surround differs from the center. Burg et al. (2021) shows that a single layer model with divisive normalization can predict the V1 response to natural signals well. This additionally also supports our claim that our model with CH divnorm is more biologically plausible. Coen-Cagli et al. (2012) provides an account of orientation-based contextual modulation in early vision by introducing a model of natural images that includes grouping and segmentation of spatially neighboring features, which is different from our channel-wise normalization solution. Cirincione et al. (2022) add divisive normalization to the front-end of VOneNet (Dapello et al., 2020) to show robustness to image corruptions; however, VOneNet uses a fixed-weight Gabor filter-bank as opposed to learning V1 feature representations. Veerabadran et al. (2022) introduced DivNormEI, which performs divisive normalization within the spatial neighborhood of each channel and applies lateral inhibition and excitation by weighted sum across channels, demonstrating improved performance in large-scale object recognition tasks.

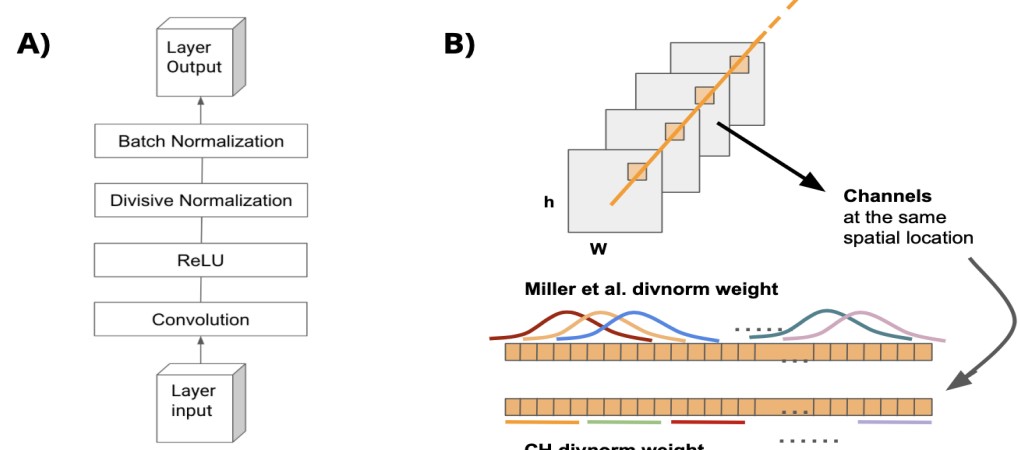

Figure 1: **Schematic. A)** A convolution module comprising of batch normalization, divisive normalization, ReLU, and a convolution operation. **B)** Comparing divisive normalization modules between Miller et al. (2022) and ours (CH)

We separate ourselves from these previous works by training an end-to-end image-computable model (i.e., one that can accept any image as input) with divisive normalization modules inserted within intermediate stages of the network. As opposed to Cirincione et al. (2022), we enforce learned local competition between neuronal sub-populations. We keep our formulation of divisive normalization as a simple computation, as emphasized by Carandini & Heeger (2012), in contrast to a relatively complex one from Miller et al. (2022). And we show the emergence of learned competition between color-opponency cell types, in addition to Gabor-like orientation selective cells, in the model's V1.

## 3 METHODOLOGY

### 3.1 MILLER ET AL. DIVISIVE NORMALIZATION FORMULATION

Miller et al. (2022)'s version of divisive normalization uses an exponentially weighted sum of the unnormalized activations of nearby channels at the same spatial position in a specific layer of a neural network:

$$R_c(x) = \frac{y_i^l(x)^2}{\left(k\left(1 + \frac{\alpha}{\lambda}\sum_{j=-4\lambda}^{4\lambda} y_{i+j}^l(x)^2 e^{-|j|/\lambda}\right)\right)^\beta} \tag{1}$$

where $\beta$, $\alpha$, $k$, and $\lambda$ are all learnable parameters, independently for each layer. The normalization window slides through all neurons in a given layer, implying that each neuron will not have a single group of neurons it competes with every time but several groups of neurons. This essentially convolutional formulation is the main reason why our implementation over discrete neighborhoods is more efficient. All trainable parameters ($\lambda$,$k$, $\alpha$, $\beta$) used in this formulation are shared among all channels and spatial positions within each layer. Appropriate padding is added to the output of the convolutional layer before divisively norming them, ensuring that neurons at the edges are adequately addressed.

### 3.2 OUR FORMULATION OF DIVISIVE NORMALIZATION

We use a form of normalization similar to Carandini & Heeger (2012)'s version for V1 contrast normalization, with $n = 2$. More specifically, let, after a certain convolutional layer $l$ of the model with $C$ channels, $y^l(x) \in \mathbb{R}^{C \times H \times W}$ represent the activity of all $C$ neurons, with $H$ and $W$ denoting the spatial dimensions of the activation map, and $x$ the input. We first break down the $C$ channels

(neurons) into $P$ non-overlapping neighborhoods, each of size $C/P$. Subscription to a neighborhood (neighborhood) defines which neurons compete with each other. Then, for each neighborhood $p \in [P]$, and for each neuron $i$ in $p$,

$$R_i = \gamma_p \cdot \frac{y_i^l(x)^2}{\sigma_p^2 + \sum_j y_j^l(x)^2} \tag{2}$$

Here, $\gamma_p$ is a multiplicative coefficient for neighborhood $p$ in layer $l$, and $\sigma_p$ is additive constant for neighborhood $p$. These are both learnable parameters. $y_i^l(x)$ is the activity of each neuron at $i^{th}$ channel after ReLU, and $j$ ranges over the neurons in the $p^{th}$ neighborhood. We assign independent learnable neighborhood-specific parameters $\lambda_p$ and $\sigma_p^2$ to each neighborhood in every layer. Independent multiplicative and additive constants are used to adjust for the possible qualitative differences between neighborhoods from the initialization process.

Henceforth, we abbreviate our formulation as *CH divisive normalization*.

### 3.3 Asymptotic Computational Complexity

For Miller et al. (2022)'s formulation of divisive normalization, a sliding window technique is used (i.e., neighborhoods are overlapping). Each neighborhood has a size of $8\lambda$, where $\lambda$ is learned during training. This means that, in the worst case, $\lambda = N/8$, with $N$ being the number of neurons in that layer. There are $(N - 8\lambda + 1)$ total neighborhoods. If, for every neighborhood, divisive normalization needs to take into account all $8\lambda$ neurons in that neighborhood, then the total number of computations becomes $(N - 8\lambda + 1) \cdot 8\lambda = \mathcal{O}(N^2)$ in the worst case (i.e., $\lambda = N/8$).

On the other hand, our formulation of divisive normalization subscribes neurons to non-overlapping neighborhoods, which means that every neuron is visited only once, making the total number of computations $\mathcal{O}(N)$.

### 3.4 Multivariate Gaussian Weight Initialization

While all of our models implement Kaiming initialization for model parameters (He et al., 2015) as the one resulting in highest categorization performance, we also implement a parameter initialization scheme to explore how local competition gets enforced among neuronal sub-populations. Local competition is usually seen between groups of neurons that perform similar computation; for example, between cell types preferring slightly varying orientations. Thus, in CH divnorm, to encourage similarity between neurons within a neighborhood while allowing for differences between neurons across separate neighborhoods, we implement a multivariate Gaussian parameter initialization. We first randomly sample a mean vector $\mu_p$ from the Kaiming distribution for each neighborhood. We then initialize filter weights within a neighborhood by sampling from a circular multivariate Gaussian distribution, with $\mu_p$ serving as the mean vector and the standard deviation set to $g_p \times$ the standard deviation of the Kaiming distribution, where $g_p$ is a hyperparameter. This initialization promotes greater similarity within each neighborhood while ensuring more diversity between neighborhoods, encouraging neurons in the same neighborhood to learn similar yet competitive features.

### 3.5 Neuroanatomical Considerations

**Divisive Normalization with other forms of normalization.** We evaluate how divisive normalization works with and without other forms of normalization like batch normalization. We observe that our formulation of Divisive Normalization in tandem with batch normalization results in a slightly higher performance than not using batch normalization, but this does not limit the practicability of our proposed method. Batch normalization helps improve performance because it (a) reduces internal covariate shifts, and (b) leads to a quicker convergence of the model (Ioffe, 2015). Divisive normalization is not like other types of normalization used in deep learning research - it does not perform standardization (zero mean, unit standard deviation) of model activations; instead, it enforces local competition for contrast gain control and sharpening of neuronal responses. Biologically, it might not be incorrect to think that several different normalization processes (regulated by biochemical pathways, somatic, dendritic, or synaptic computation, or modulated by astrocytes etc.)

Table 1: **Performance of a two-layer CNN on CIFAR-10 and AlexNet on CIFAR-100.** Results are shown as the mean and standard deviation over 10 runs of random model initialization seeds. Best result is in bold; second-to-best, underlined.

| Index | Activation/Normalization | Top-1 accuracy |
|---|---|---|
| | *Two-layered CNN evaluated on CIFAR-10* | |
| 1 | ReLU + BatchNorm | $69.50 \pm 0.23$ |
| 2 | ReLU + CH DivNorm + BatchNorm | $\underline{70.53 \pm 0.00}$ |
| 3 | ReLU + CH DivNorm | $63.34 \pm 0.19$ |
| 4 | CH DivNorm + BatchNorm | $\mathbf{73.07 \pm 0.01}$ |
| | *AlexNet evaluated on CIFAR-100* | |
| 5 | ReLU + BatchNorm | $57.9 \pm 0.28$ |
| 6 | ReLU + Miller et al. DivNorm + BatchNorm | $58.6 \pm 0.16$ |
| 7 | ReLU + CH DivNorm + BatchNorm | $\underline{60.6 \pm 0.30}$ |
| 8 | ReLU + CH DivNorm | $51.6 \pm 0.24$ |
| 9 | CH DivNorm + BatchNorm | $\mathbf{62.7 \pm 0.34}$ |

occur over neuronal activities. For example, while divisive normalization is a canonical neural computation that leads to local enforcement of competition among similar functional neurons (Carandini & Heeger, 2012), homeostatic mechanisms can prevent neural activity from being driven towards runaway activity or quiescence; one such homeostatic mechanism is the adjustment of synaptic excitability so that firing rates remain relatively constant (Turrigiano & Nelson, 2004). Moreover, it is not the case that using two different normalization techniques leads to a significant increase in computational expense.

**Placement of divisive normalization layers in the network.** Divisive Normalization is not only known to explain responses in the primary visual cortex (V1), but also seen to operate in a variety of other regions of the visual system: light adaptation in the retina, contrast normalization in the retina and lateral geniculate nucleus (LGN), and visual processing in higher visual cortical areas beyond V1 (Carandini & Heeger, 2012). This prompts us to use the divisive normalization module throughout the network.

## 4 EXPERIMENTAL RESULTS

### 4.1 SHALLOW NETWORKS WITH DIVISIVE NORMALIZATION DO NOT REQUIRE HALF-WAVE RECTIFICATION

Divisive normalization performs a non-linear computation over the raw activity of neurons based on the activity of nearby neurons in their neighborhood. We thus test the ability of divisive normalization to serve as an effective module that performs the roles of both an activation function and activity modulation in neural networks. To this end, we first compose a neural network with two convolutional layers. We embed four different combinations of activations functions within this network based on the use of half-wave rectification (such as ReLU), divisive normalization, and batch normalization. We also evaluate the same combinations of activation functions in AlexNet. We summarize our results in Table 1.

The best categorization performance on both the CIFAR-10 and CIFAR-100 datasets (Krizhevsky & Hinton, 2009)—that consists of tiny real world visual stimuli for 10- and 100-way categorization—is achieved by the model that only performs divisive and batch normalization. Interestingly, half-wave rectification as an activation function does not seem to be required prior to the computation of divisive normalization. Half-wave rectification only lets excitatory neuronal responses to pass through. On the other hand, divisive normalization squares the raw neuronal activity, and thus lets both excitatory and inhibitory signals pass through in the absence of being preceded by half-wave rectification. This version of divisive normalization relies purely on signal strength (i.e., the magnitude of the responses), and introduces local competition between neuronal sub-populations to reduce signal redundancy. As opposed to half-wave rectification that removes inhibitory signals, divisive normalization learns to remove those signals (whether excitatory or inhibitory) that lead to

Table 2: **Performance of AlexNet on ImageNet.** We use model variations to evaluate categorization performance, runtime, GPU memory, and the number of extra parameters due to divisive normalization for a specific layer over a baseline model required during training. All models use batch normalization after every convolutional layer. Results are shown as the mean and standard deviation over 5 runs of random model initialization seeds. Best results are shown in bold; second-to-best, underlined. Here, $P$ represents the number of neighborhoods in a given layer of the model. Top-1 accuracy for Miller et al. (2022) is taken from their paper.

| Model Variant | Top-1 Accuracy | Mean Runtime (sec/epoch) | GPU Memory (G) | # Extra Parameters |
|---|---|---|---|---|
| Baseline | $58.5 \pm .10$ | 970 | 1.4 | 0 |
| Miller et al. (2022) DivNorm + ReLU | $\underline{61.3 \pm .00}$ | 2800 | 2.9 | 4 |
| CH DivNorm + ReLU (Shared CH para.) | $61.2 \pm .10$ | 1210 | 1.9 | 2 |
| CH DivNorm + ReLU | $\underline{61.3 \pm .12}$ | 1280 | 1.9 | $2 \times P$ |
| CH DivNorm | $\mathbf{62.3 \pm .08}$ | 1250 | 1.9 | $2 \times P$ |

Table 3: **Accuracy and Runtime of VGG-16 on ImageNet.** Results are shown as the mean and standard deviation over 5 runs of random model initialization seeds.

| Model Variant | Top-1 Accuracy | Top-5 Accuracy | Runtime (sec/epoch) |
|---|---|---|---|
| Baseline | $70.8 \pm .09$ | $90.0 \pm .18$ | 5150 |
| CH DivNorm + ReLU | $72.7 \pm .08$ | $90.6 \pm .12$ | 7486 |
| CH DivNorm | — | — | — |
| Miller et al. (2022) DivNorm | — | — | — |

feature redundancy. This is particularly helpful where each neuron must encode the most relevant features to discriminate between categories.

Since the output from the divisive normalization computation is all positive neuronal responses, adding batch normalization helps stabilize learning and ensure better gradient flow by standardizing responses to have zero mean and unit standard deviation. This leads to improved categorization performance, both in the absence and presence of ReLU.

## 4.2 DIVISIVE NORMALIZATION IMPROVES CATEGORIZATION PERFORMANCE

We demonstrated in the previous section that divisive normalization leads to improved categorization performance for two different deep neural networks, on two separate datasets. Next, we more rigorously quantify the improvement by evaluating on the ImageNet dataset (Deng et al., 2009).

Overall, categorization performance improves for all model variants that use divisive normalization (see Tables 2, 3). For AlexNet, Miller et al. (2022) use four parameters ($\lambda$, $\alpha$, $\beta$, and $\kappa$) for divisive normalization that are learned for every layer of the network. Due to the way that they implement this computation, they incur a significantly large runtime that is approximately three times that of the base model. There is also an approximately twice GPU memory requirement over the base model. These overheads come with an improved performance on categorization. For our formulation of divisive normalization, we experiment with the number of divisive normalization parameters per layer of the network. Any parameter choice results in an improved performance over the base model. If we fix $\lambda$ and $\sigma$ to the same value for all neighborhoods within a layer, we observe slightly, but not significantly, drop in performance compared to Miller et al. (2022). Learning these two parameters for each neighborhood in a layer improves categorization performance, while incurring a modest increase in the runtime and GPU memory. Learning separate parameters for each neighborhood helps the divisive normalization computation to capture heterogeneous neural dynamics across neuronal sub-populations.

To assure the generalization of our formulation of divisive normalization to deeper models—an important criterion that Miller et al. (2022) do not explore—we applied it to the VGG-16 (Simonyan & Zisserman, 2015) model class. Unfortunately, we do not have results for when we implement VGG-16 with Miller et al. (2022)'s formulation of divisive normalization with or without ReLU because it turns out to be very computationally expensive and does not converge successfully. As shown in Table 3, the use of divisive normalization outperforms the baseline on ImageNet. However, given

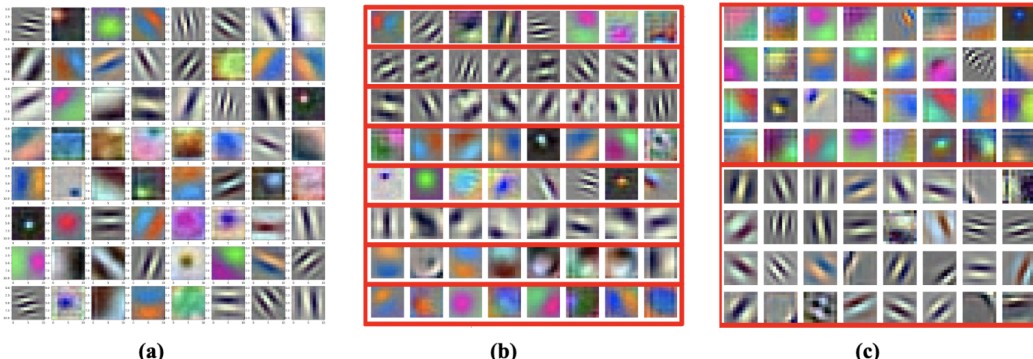

(a)                                (b)                                (c)

Figure 2: **Learned filters for different models.** First convolution layer filters for **(a)** baseline AlexNet, **(b)** AlexNet with divisive normalization (neighborhood size = 8), and **(c)** AlexNet with DN (neighborhood size = 32). There is one neighborhood per red box for **(b)** and **(c)**. We find local competition emerges between Gabor-like orientation selective filters as well as color-opponent cell types in **(b)** and **(c)**

that this model is deeper than previously explored model classes like AlexNet, using only divisive normalization without being preceded by half-wave rectification led to the training not converging. Hence, it appears that what limits the application of divisive normalization as comprising an activation function is dependent on the depth of the model, although a more thorough exploration of hyperparameters is necessary to verify this speculation.

### 4.3 DIVISIVE NORMALIZATION ENFORCES LOCAL COMPETITION BETWEEN GABOR AND COLOR-OPPONENT CELLS, AND SHARPENS RESPONSES

Figure 2 shows features learned by the neurons in first convolutional layer of AlexNet with and without divisive normalization. AlexNet, as the authors of the paper already demonstrate, learns orientation-selective and color-opponent simple cells in its first convolutional layer (Krizhevsky et al., 2012). There is, however, no neuroanatomical structure that is tied to the functional roles that these cell types play in visual processing (figure 2a). Gabor-like orientation-selective cells are thought to compete with each other for enhanced edge perception and contrast normalization, while color-opponent cells are thought to compete with each other for refining color perception. When we introduce divisive normalization to form neighborhoods of neuronal sub-populations over which local competition is enforced, the model learns to make this competition between similar cell types, an emergent property that is apparent in Figure 2b, but is most obvious in (Figure 2c). Again this behavior emerges from the model; while previous models have shown Gabor filter-like competition, we believe this is the first evidence of color opponency emerging from divisive normalization.

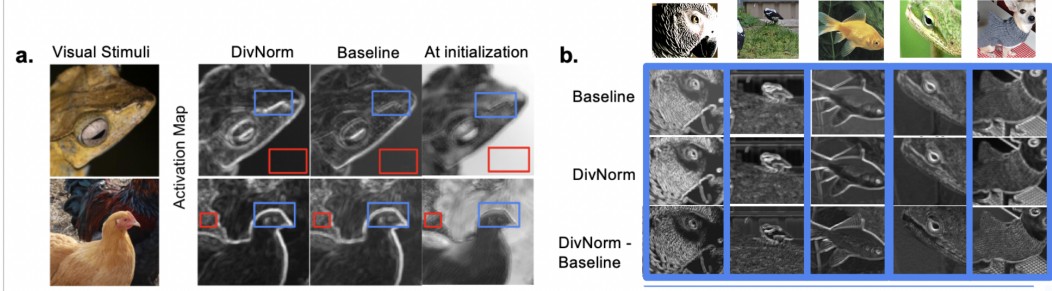

Figure 3: **Activation Maps for the first convolutional layer of AlexNet.** We create activation maps for different visual stimuli for AlexNet with and without divisive normalization. In **(b)**, we also visualize the difference in activation maps between the models. The color map for this difference is re-scaled to make differences more visible. Brighter pixels represent strong activation. Responses on these visual stimuli are representative of those on other stimuli from the dataset.

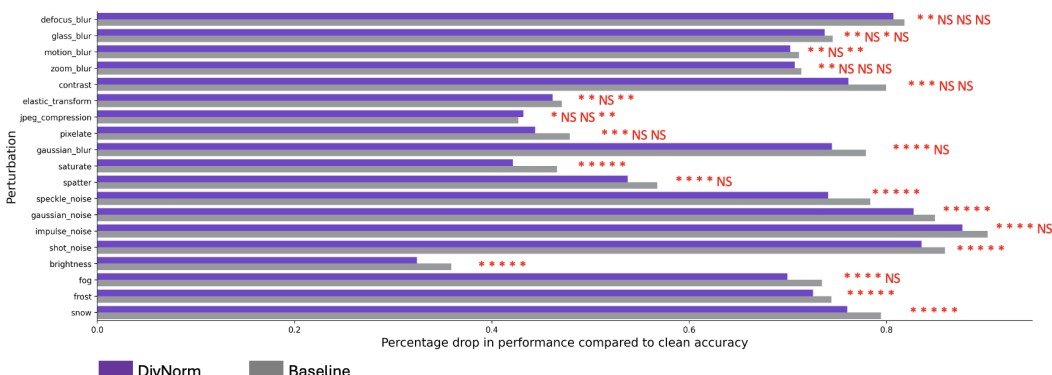

Figure 4: **Percentage drop in performance of AlexNet on perturbed images compared to clean accuracy (on unperturbed images)**. Longer bars mean that the model is less robust to this particular noise. The purple and gray lines show the average percentage drop over the five noise levels. To the right of the lines, we show the results of significance testing for each noise level individually (averaged over four runs). We apply a t-test to the percent reduction in performance $\left( \frac{\text{clean accuracy} - \text{noise accuracy}}{\text{clean accuracy}} \right)$ averaged over the four runs for models trained with and without divisive normalization. Asterisks refer to significant differences within each noise level using a t-test. Hence, for example, * * NS NS NS means that for noise levels 1 and 2, we find $p < 0.05$, and for noise levels 3, 4, and 5, there is no significant difference.

Furthermore, we identify the role of divisive normalization in sharpening of neuronal responses through activation maps for different visual stimuli (figure 3). Looking at the activation maps, it appears that divisive normalization leads to sharpened edge detection (as indicated by a more prominent and intense color on the activation map) than the base model. Both the base model and an unoptimized model show some activity in the background of the object (as indicated by relatively brighter colors than for divisive normalization). To make such differences more prominent, we additionally visualize the difference in activation maps between the models trained with and without divisive normalization. Stronger visible responses in the difference map are attributable to the sharpening effect that divisive normalization demonstrates. Darker patches here, on the other hand, signal suppressed activity from having reduced redundancy between responses—this redundancy might either stem from noise or unwanted variability that needs to be removed for more robust feature learning.

## 4.4 DIVISIVE NORMALIZATION IMPROVES ROBUSTNESS TO LOCALIZED IMAGE CORRUPTIONS

Finally, we use the ImageNet-C benchmark dataset (Hendrycks & Dietterich, 2019) to test the robustness of using our formulation of divisive normalization on corrupted (i.e., perturbed) images. This dataset introduces algorithmically-generated parametric distortions to images from the ImageNet test set to evaluate model robustness. We visualize performance on this dataset in Figure 4. Across a variety of different perturbations and noise levels, divisive normalization helps in statistically significant reduction in performance drop from the clean accuracy on unperturbed images compared to the base model. However, for some perturbation classes like defocus and zoom blurring, divisive normalization does not significantly help under high noise levels. Overall, divisive normalization seems to help significantly when facing color or pixel-wise variations in images (such as brightness, saturation, and gaussian noise) but not for spatially-induced distortions (such as blurring and pixel compression). This makes sense, since divisive normalization primarily focuses on normalizing neuronal responses within local receptive fields, why may not adequately address spatially global distortions.

## 5 DISCUSSION

In this work, we presented a lightweight, memory and runtime efficient implementation of divisive normalization that is inspired by Carandini & Heeger (2012). We analyzed its role in learning unsupervised local competition between neuronal sub-populations—orientation-selective and color-

opponent cell types. Divisive normalization leads to improved performance on categorization, and makes the model more robust to localized parametric distortions. Finally, we show that divisive normalization can be used as a module that acts as both an activation function and modulates responses, not requiring the use of a half-wave rectifier in shallower networks.

We acknowledge certain limitations in the work that we present. First, we limit our analyses to AlexNet, VGG-16, and a two-layer CNN. While these are more anatomically consistent than very deep networks like ResNet-50, since there are somewhere on the order of 15-20 visual areas in humans (Van Essen, 2003), it will still be interesting to understand how very deep networks perform with divisive normalization. In this work, we were constrained by computational power from performing this more demanding experiment. Another avenue for improvement is quantifying learned filter similarity for models trained with divisive normalization. Although we visually examined the grouping patterns of different cell types (orientation-selective and color-opponent) in the first convolutional layer of AlexNet (in section 4.3), we lacked a robust metric to quantify this similarity. Thirdly, using Brain-Score (Schrimpf et al., 2018; 2020) as a set of metrics to analyze the capabilities of different layers of a model trained with divisive normalization to predict real neuron responses in the biological ventral visual cortex's V1, V2, V4, and the inferior temporal cortex (IT) would be a strong test of such a canonical neural computation.

Finally, there are some shortcomings in the way we implement divisive normalization. Miller et al. (2022) define overlapping neighborhoods for neurons to be divisively normed over (with exponentially decaying weights away from the center), while we define neighborhoods to be non-overlapping (neurons only affect other neurons performing similar computation and in their neighborhood). Both of these formulations are of value in their own right (similar to requiring both functional segregation and coherent perception). We could create a more biologically plausible formulation of divisive normalization that would incorporate the contribution of recurrent amplification from higher visual areas, i.e., amplifying weak inputs more than strong inputs (Heeger & Zemlianova, 2020). Additionally, it would be interesting to understand what effects divisive normalization has in vision transformers (Vaswani, 2017), which have become state-of-the-art for image categorization tasks. More specifically, divisive normalization can be used to explain how responses in the visual cortex are modulated by attention, wherein attention multiplicatively enhances the stimulus drive before normalization (Carandini & Heeger, 2012). Performing divisive normalization over the attention weights of the transformer encoding layers might reveal new insights and serve as a good direction for future work.

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

## A    CODE AVAILABILITY

We make the code available at this anonymous repository: https://github.com/anonymoustech1234/Divisive_Normalization.

## B    TRAINING DETAILS

All experiments are run on a single GPU with 24 GB of GPU memory, 8 CPUs (8 number of workers), and 14 GB of computer memory (RAM) on an internal cluster.

**Two-Layered Model on CIFAR-10.** For the CIFAR-10 (Krizhevsky & Hinton, 2009) dataset, we implemented a two layer model with the first layer as `torch.nn.Conv2d(in_channels=3, out_channels=32, kernerl_size=3, stride=1, padding=1)` and the second layer as `torch.nn.Conv2d(in_channels=32, out_channels=64, kernel_size=3, stride=1, padding=1)`. The model is trained using stochastic gradient descent, with the learning rate initialized to 0.01, imposing a weight decay of 0.0001 and a momentum of 0.9.

**AlexNet on CIFAR-100.** Since stimulus size for CIFAR-100 is relatively small, we change the first two convolutional layers of AlexNet to have a kernel size of 3. The model is trained for 90 epochs with stochastic gradient descent and a batch size of 256. The learning rate is initialized to 0.1, and decays by a factor of 0.1 every 30 epochs. We impose a weight decay of 0.0001 and a momentum of 0.9. We initialize divisive normalization parameters $\lambda$ to 10 and $\sigma$ to 0.1.

**AlexNet on ImageNet.** This model is trained with stochastic gradient descent using a momentum of 0.9, a weight decay of 0.0001, a batch size of 128, and a plateau learning rate scheduler with factor of 0.1 and patience of 5. The learning rate was initialized to 0.01. We again initialize divisive normalization parameters $\lambda$ to 10 and $\sigma$ to 0.1.

**VGG-16 on ImageNet.** The model is trained with stochastic gradient descent using a momentum of 0.9, a weight decay of 0.0005, a batch size of 64, and a step learning rate scheduler with a factor of 0.1 and step size of 15. The learning rate is initalized to 0.01.

For all experiments discussed, the neighborhood size for divisive normalization for the first convolutional layer is fixed at 8 filters per neighborhood.

## C    Forming Neighborhoods Within Each Layer

For enforcing local competition between neuron sub-populations, to then to perform divisive normalization on them, such sub-populations (or neighborhoods) must first be created. We do so as follows:

We use $p$ as a hyperparameter that trades off the size and the number of neighborhoods in each layer of the network. Let $a$ denote the neighborhood size, $b$ the number of neighborhoods, and $c$ the total number of neurons (i.e., the output channels) in a specific layer. We use the expressions:

$$a = 2^{\mathrm{int}(p \log_2(c))} \tag{3}$$
$$b = \mathrm{floor}(c/a) \tag{4}$$

In short, $p \in [0, 1]$. The larger $p$ is, the larger the neighborhood size, and consequently fewer number of neighborhoods in a model layer. Table 4 computes these statistics for each layer of AlexNet when $p = 0.5$, the value used for all our experiments. Our analyses show that there is no significant difference in performance when changing $p$ between 0.3 to 0.7. Performance, however, drops significantly when $p$ approaches either 0 or 1.

Table 4: Neighborhood size and number of neighborhoods in AlexNet when $p = 0.5$.

| Layer | Total Filters | Neighborhood Size | Number of Neighborhoods |
|---|---|---|---|
| First Layer | 64 | 8 | 8 |
| Second Layer | 192 | 8 | 24 |
| Third Layer | 384 | 16 | 24 |
| Fourth Layer | 256 | 16 | 16 |
| Fifth Layer | 256 | 16 | 16 |

## D    Multivariate Gaussian Weight Initialization

In Section 3.4, we proposed using Multivariate Gaussian initialization to encourage similarity and competition within neighborhoods.

Fig 5 visualizes filters initialized with Kaiming initialization compared to Multivariate Gaussian initialization. In the right image (Multivariate Gaussian initialization), filters within each neighborhood are much more similar to each other than across neighborhoods. Each red box marks a neighborhood. For comparison in this section, we used the same initializations for both the baseline AlexNet and AlexNet with CH DivNorm.

Fig 6 shows the filters after training for AlexNet with CH DivNorm. Each row represents a neighborhood in which neurons are divisively normalized together, thus competing with each other. Each red box marks a neighborhood with filters that are similar but differ in the orientation or color they capture. Filters trained with Multivariate Gaussian initialization genuinely contain more such neighborhoods.

Fig 7 shows the filters after training for baseline AlexNet. Without divisive normalization, no clear grouping is observed among the filters. The Gabor-like filters learned vary in frequency and scale and do not form a complete set. There is no noticeable difference in the patterns learned with Kaiming or Multivariate Gaussian initialization

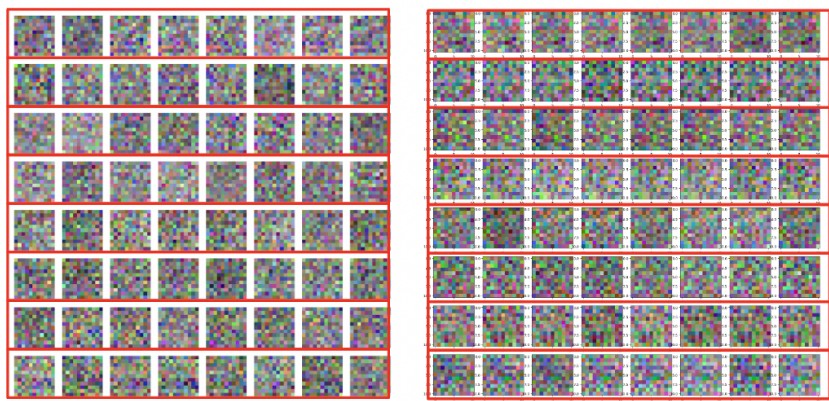

Figure 5: Comparison of Filters Initialized with Kaiming Initialization (left) vs. Multivariate Gaussian Initialization (right), using Means and Standard Deviations Sampled from Kaiming Initialization

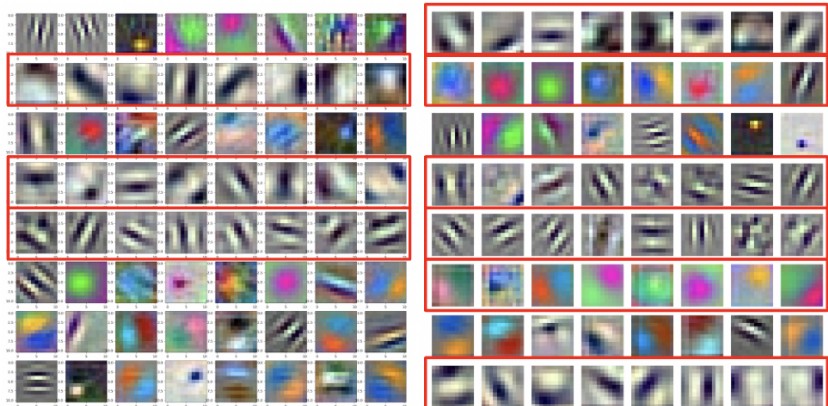

Figure 6: Comparison of Filters in AlexNet with CH DivNorm After Training: Kaiming Initialization (left) vs. Multivariate Gaussian Initialization (right)

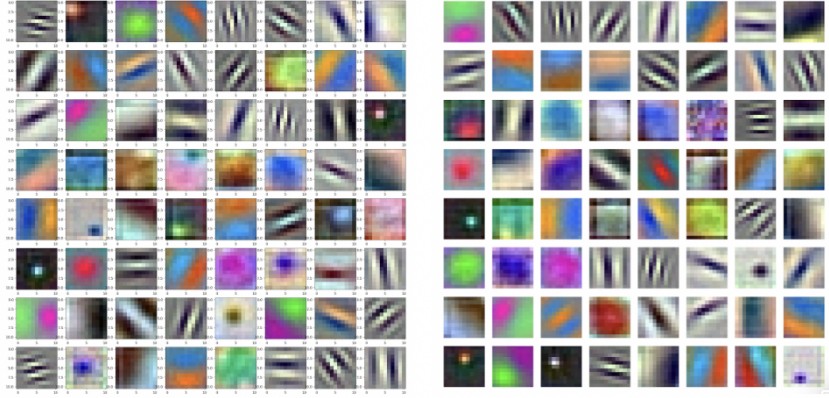

Figure 7: Comparison of Filters in AlexNet Baseline (No DivNorm) After Training: Kaiming Initialization (left) vs. Multivariate Gaussian Initialization (right)

# E ACCURACY ON CORRUPTED IMAGES

In section 4.4, we show the average performance drop of AlexNet with and without divisive normalization compared to the clean accuracy. In figure 8, we provide the raw accuracies for each perturbation class and noise level.

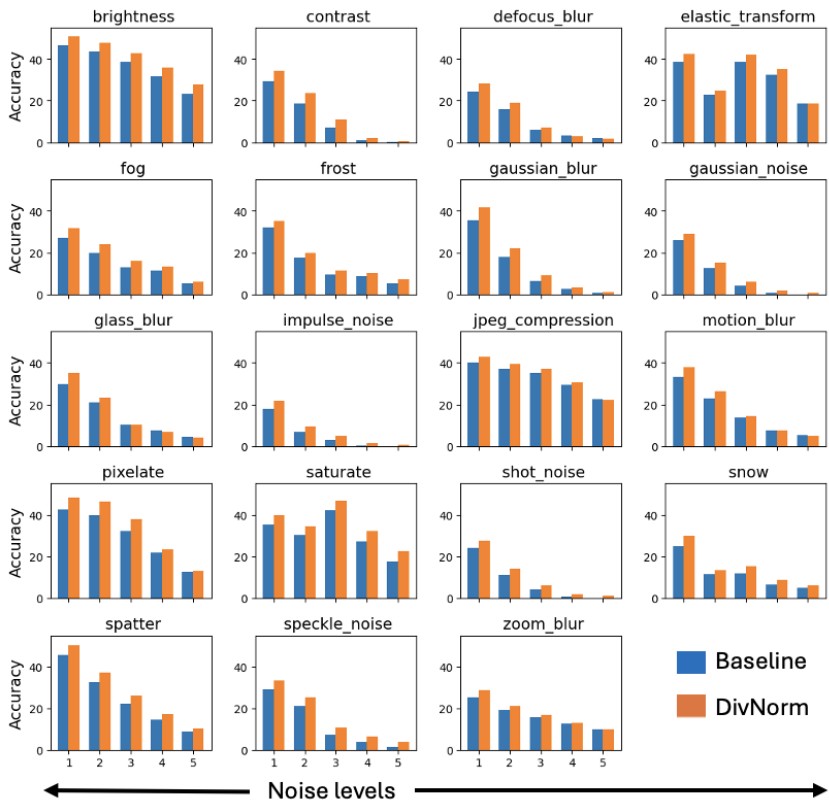

Figure 8: Top-1 performance of AlexNet on different perturbation classes and noise levels.

