# OpenReview forum: "Modeling Divisive Normalization as Learned Local Competition in Visual Cortex"
_ICLR.cc/2025/Conference — Submitted to ICLR 2025_

### Official Review · Reviewer_fbkc · 2024-11-01

**Soundness:** 1
**Presentation:** 2
**Contribution:** 1
**Rating:** 1
**Confidence:** 4

**Summary:**

The paper introduces an “group-wise” implementation of divisive normalization for convolutional neural networks (CNNs), inspired by biological visual processing. Divisive normalization is a common computation in the visual cortex where a neuron's response is modulated by the activity of neighboring neurons. The authors propose a simple formulation that divides neurons into non-overlapping neighborhoods and normalizes their responses within each group. Unlike previous implementations that use overlapping neighborhoods, their approach is more efficient while maintaining or improving performance. The authors also introduce a novel parameter initialization scheme using multivariate Gaussian distributions to encourage similarity between neurons within the same neighborhood.

The authors perform experiments on various architectures (AlexNet, VGG-16, and a two-layer CNN) and datasets (CIFAR-10, CIFAR-100, and ImageNet). First, compared to baseline models, their divisive normalization improves classification accuracy and makes models more robust to image corruptions. Second, in shallow networks, divisive normalization can  replace traditional activation functions like ReLU. Third, they show that their approach leads to the emergence of biologically plausible competition patterns between both orientation-selective and color-opponent cells in the network's first layer.

**Strengths:**

- The paper is generally well written.
- The paper addresses an important question: The benefit and computational role of divisive normalization which is observed throughout the brain.

**Weaknesses:**

My biggest concern is about the contribution of the paper. In my view it could be two things:

1. Divisive normalization as a deep learning mechanism: The paper could show that divisive normalization improves the performance of deep networks in a relevant way.
2. Normative interpretation of divisive normalization to better understand neuronal coding.

However, the paper does not deliver for either.

Regarding 1, they use architectures that are substantially below current state of the art. The best performing architecture on ImageNet in the paper achieves a performance of 62.3% top-1 accuracy, improving about 3% over their baseline. This improvement is 10 times smaller than the 30% difference between their best network and the current state of the art of an OmniVec ViT with 92.4% top one accuracy. Thus, I do not see how this paper makes a contribution to current deep learning research.

Regarding 2, there has been so much work on normative aspect of divisive normalization before that, some of which is cited some not (including, for instance, a Nature Neuroscience paper by Odelia Schwarz and Eero Simoncelli; https://www.nature.com/articles/nn0801_819). In comparison to that the results on the neuroscience side are also small or not novel. For instance, the “color opponency” emphasized by the authors can also be found by simple linear ICA. For that see:

- Lee, T. W., Wachtler, T., & Sejnowski, T. J. (2000, May). The spectral independent components of natural scenes. In *International Workshop on Biologically Motivated Computer Vision* (pp. 527-534). Berlin, Heidelberg: Springer Berlin Heidelberg.
- Eichhorn, J., Sinz, F., & Bethge, M. (2009). Natural image coding in V1: how much use is orientation selectivity?. *PLoS computational biology*, *5*(4), e1000336.

So, in summary I cannot see the contribution of this paper. To address this the authors will need to show what gap of knowledge they address and where their research provides a significant contribution to the field.

Finally, I find some of the statements in the paper a bit to strong or wrong. One example for “too strong” is:

> The superiority of CNNs over fully-connected networks can be attributed to their biologically-inspired properties of shared filter weights, translation invariance, and larger receptive fields with depth.
>

Another example for “too strong” is

> We separate ourselves from these previous works by training an end-to-end image-computable model (i.e., one that can accept any image as input) with divisive normalization modules inserted within intermediate stages of the network.
>

Burg et al. also train deep network end to end, as do Ren et al. The training targets might be different, but they are not the first to do this.

An example for “wrong” is:

> The numerator of their normalization formula is still first order, so it cannot remove the second order dependency.
>

I do not agree with this statement. First of all, most people would understand second order dependency as “covariance” or “correlation”. This can easily be removed by a linear operation like PCA. However, you probably meant higher that second order dependencies, i.e. variance correlations. However, some of them can also be removed by a linear numerator if the denominator is nonlinear. Relevant work for this is

- F. Sinz, M. Bethge The Conjoint Effect of Divisive Normalization and Orientation Selectivity on Redundancy Reduction Advances in Neural Information Processing Systems 21, 1521-1528
- F. Sinz, E. P. Simoncelli, M. Bethge Hierarchical Modeling of Local Image Features through Lp-Nested Symmetric Distributions Advances in Neural Information Processing Systems 22, 1696-1704
- Lyu, Siwei, and Eero P. Simoncelli. "Nonlinear extraction of independent components of natural images using radial gaussianization." *Neural computation* 21.6 (2009): 1485-1519.
- Lyu, Siwei “Divisive Normalization: Justification and Effectiveness as Efficient Coding Transform”, Advances in Neural Information Processing Systems 23 (NIPS 2010)

**Minor (do not expect you to respond to this)**

- Figure 4: x-axis is in fraction, not percent.

**Questions:**

Given the weaknesses above, I really only have two question:

- What is gap of knowledge your paper addresses and where does your research provide a significant contribution to the field?
- What sets your work apart from previous work on divisive normalization in deep network or normative models of neuronal coding in V1?

---

### Official Review · Reviewer_9T5q · 2024-11-03

**Soundness:** 3
**Presentation:** 2
**Contribution:** 2
**Rating:** 6
**Confidence:** 4

**Summary:**

Inspired by visual neuroscience literature, the authors used a simple but elegant implementation of divisive normalization similar to group norm. They found using this module smaller networks can work well without ReLU layers, and deeper network can improve their categorization performance while improve on robustness to pixel wise distortions. Further they found color opponency & orientation tuning (gabor filter) cells self organized into different module.

**Strengths:**

- Their implementation of div norm is simple and lightweight, close to what is used in ML (group norm) and neuro literature.
- The authors performed relatively thorough evaluation of their method on small scale to AlexNet, VGG networks.
- Networks using div norm + batch norm can get rid of their ReLU nonlinearity is a though provoking result, which I didn’t know about before.
- “Comparing *Divisive Normalization with other forms of normalization*” discussion is indeed interesting and important. seems like feature competition in divnorm and the standardization effect of batch norm can serve different computational purposes.

**Weaknesses:**

- The asymptotics complexity computation in Sec 3.3 seems not correct: when *λ = N/8, (N − 8λ + 1) * 8λ = N, which is not O(N^2).* I feel the logic of the aymptotic analysis didn’t follow well. The authors should check.
     - Maybe the authors can provide a more step-by-step derivation of their complexity analysis, or clarify any assumptions they made that led to their O(N^2) conclusion. This would help pinpoint where the discrepancy might be occurring.
- Scaling the results to deeper, more modern networks (e.g. ResNet50) is lacking.
     - Maybe the authors could discuss any specific challenges they anticipate in applying their method to ResNet50 or other modern architectures. Additionally, are there some specific experiments that can demonstrate the scalability of their approach to deeper networks?

**Questions:**

- **Main question:**
I think your divnorm layer can be thought of as a composition of a square activation function ($\phi(x)=x^2$), and a simpler divisive normalization ($R_i = \gamma_p \cdot \frac{\phi(x)_i}{\sigma_p^2 + \sum_j \phi(x)_j}$). so in some sense you also have a nonnegative, non-saturating nonlinearity implicitly in the normalization layer, which made the results about relu being unnecessary less surprising.
Given that, is there a fairer comparison, just keep the form of divisive normalization as I put above, but just change the non-negative, nonlinear function $\phi$ from squaring to ReLU, softplus, exp or so and then benchmark these models, that could be informative and a fairer evaluation of the statement that Relu is unnecessary for batchnorm + divnorm layers.
- I really like the result of div norm lead to similar tuning cell types self-organize into one module.
My question is, does that need to be div norm in each group? or group norm will have similar effect, can you check?
- Many modern networks are trained with group norm in place, could this be a in place drop in for group norm?
- The current related work section is well written but text only. It will be helpful to put the related works’ implementation of divisive normalization into comparable notations / a table, I feel organizing that comparison is already a big contribution to the field.

---

### Official Review · Reviewer_QNC1 · 2024-11-04

**Soundness:** 3
**Presentation:** 3
**Contribution:** 1
**Rating:** 3
**Confidence:** 4

**Summary:**

The paper proposes divisive normalization as a canonical computation to improve convnet's performance on image recognition. The authors conduct experiments on small-scale visual recognition tasks such as CIFAR and the classical AlexNet architecture on ImageNet. They find small improvements in classification accuracy of around 2%. In addition, they rediscover the finding from the original AlexNet that local competition separates the first layer's filters into edge detectors and color opponent units.

**Strengths:**

+ Paper is well written and easy to follow
+ Shows some improvements on AlexNet on a variety of datasets

**Weaknesses:**

- Very limited novelty
- Small improvements on networks that are far from state of the art
- Quantitative comparisons to other normalization approaches missing

**Questions:**

While the idea is simple and the paper relatively straightforward to read, I have a number of serious concerns regarding novelty, effect sizes and potential impact of the paper:

 1. Why is the only comparison you show the divisive normalization approach by Miller? How about the original LRN from AlexNet, plain batch norm, layer norm, group norm? Many of them are known to improve performance over plain convnets quite a bit, perhaps even more than your specific version of DN.

 1. As far as I remember, the original AlexNet was partitioned onto two GPUs, which had independent normalizers and the original AlexNet already resulted in the separation between edge filters and color opponent filters as you show them in Fig. 2c. What's new in your analysis?

 1. Your DN – while being more efficient than Miller's method – still incurs a quite substantial computational overhead (ca. 25% for AlexNet and 50% for VGG; Tables 2+3). Have you tested how much of an improvement you get if you simply make the networks deeper or wider. On ImageNet, neither AlexNet nor VGG are anywhere close to having enough capacity, so adding expressive power to the networks is expected to improve performance. It is thus possible (and quite likely IMHO) that you gain in performance has nothing to do with DN per se, but is simply an effect of having a more expressive network.

 1. Related to the previous point: While the classical networks provide a good proof of principle, for the method to actually gain widespread adoption it would be necessary to show that it helps on more recent, state-of-the-art architectures. For instance, where does it put the EfficientNet family of networks on the compute–accuracy tradeoff when you add DN?


### Minor:

 1. Are the activation maps in Fig. 3 normalized somehow? To me, the differences between baseline and DN look quite imperceptible and it simply looks like the DN ones have slightly more contrast. But that's somewhat arbitrary unless you somehow fix the scale.

---

### Official Review · Reviewer_eMC7 · 2024-11-04

**Soundness:** 2
**Presentation:** 3
**Contribution:** 1
**Rating:** 3
**Confidence:** 4

**Summary:**

The authors incorporate a specific form of local divisive normalization (inspired from computational models of the biological early visual system) in CNNs to evaluate whether this non-linearity leads to better performance of CNNs on object recognition and robust recognition tasks. The authors use an implementation of the traditional divisive normalization proposed by Carandini and Heeger and show that when integrated into three different networks ( two-layer CNN, AlexNet and VGG-16), these networks perform better at ImageNet recognition. This implementation of divisive normalization is described to be more efficient (compute-wise) than recent prior work (Miller et al) and is shown to additionally provide some improvements in robust recognition (evaluated via ImageNet-C). The authors also provide some analysis of the learned filters to interpret the differences compared with the baseline.

**Strengths:**

The paper is well-written and easy to understand. I believe the topic is also a useful and interesting as we know that local divisive normalization is a canonical computation in visual cortex yet it is usually replaced by simpler non-linearities in current deep learning networks. Therefore, if there was a use-case for these non-linearities at scale, it would be very interesting to the field. Additionally, relative to the prior work of Miller et al, this model seems to improve significantly (at least in the limited experiments that are shown). The additional robust recognition results are interesting and appreciated as they provide more evaluations.

**Weaknesses:**

1. I believe the **scope of experimentation is quite limited** and the lack of additional baselines makes it hard to interpret the true value of these results. The specific issues i have are:
- The authors only compare to a standard baseline, variants of their method (with CH-divnorm) and the Miller et al. divisive normalization. However, there are other prior works that have shown significant improvements over baseline with divisive normalization such as [1]. In Ren et al, in fact, they find that simple modifications of BatchNorm and LocalNorm give large improvements over their baseline BN/LN formulations. The gains they find over standard BN in similar object recognition are comparable to what is shown by CH-divnorm, therefore, I'm not sure what the practical advantage might be of using CH-divnorm vs. what is proposed in Ren. The authors (of the current submission) suggest in their related work that the Ren formulation cannot remove second-order dependencies and suggest that CH-divnorm is more biologically plausible; however, I find this line of argument weak as there are other ways to remove statistical dependencies and it is not explicitly shown that CH-divnorm would outperform the Ren et al. formulation on this set of evaluations.
- The authors claim that they only test on AlexNet, VGG-16, and a two layer CNN due to the better alignment of these architectures with biology; however, I find this argument extremely weak. Some of the best models on BrainScore are in fact very deep networks that are predictive of neural responses in visual cortex. This is because feed-forward networks can't capture the complexity of recurrence and thus require more layers to acheive similar computational capacity. If the authors are shooting for biological plausibility (which they mention many times), then at the very least they should work with architectures like the CorNet [2] which attempt to better align with visual cortex. Otherwise these set of experiments are neither very close to biology nor good benchmarks for potentially scaling to SoTA networks like ResNets or ViTs.
- I understand that compute may be a concern with training networks (and thus can give a pass to the authors in not training more networks) but evaluation is something that is quite easy to do with limited compute and the authors use only a few limited evals from object recognition (CIFAR, ImageNet, ImageNet-C). I believe these results are quite limited because the overall training category distribution is still similar between the train and test sets in these evals. Therefore, it is quite possible that the CH-divnorm parameters are tuned to statistics of these image families and do not generalize in a transfer learning setting. For example, do you see gains from CH-divnorm when evaluating linear classification performance on other transfer tasks like DTD (textures), Cars/Birds etc. (fine grained classification) and other OOD evals that are commonly used for transfer learning?
- For ImageNet-C I don't understand why the authors only evaluate their AlexNet model. Do these results also hold with VGG-16? At minimum we would like to see the robustness results transfer across these two architectures but it is unclear from the text whether this is true.
- For the ImageNet-C robustness results, while I greatly appreciate the use of statistical significance, I would like to know what these results look like when averaging across the corruption severity levels (as is done with the canonical ImageNet-C benchmark). The reason being, from a practical perspective, being robust to corruptions just at a specific severity level is not useful. However, if the networks are on average more robust across severity levels this would be a stronger indication. At the very least, even if the authors do not feel like averaging over severity levels, I would like to see the average performance gain for a given severity level, across all corruption types. Are the gains statistically significant when averaging over the corruption types?

2. The compute efficiency (which is emphasized in this work multiple times) is perplexing to me as it is never justified whether the performance gains are in fact worth the increase in compute. For example, in Table 3, we see that per epoch, the VGG model with the CH-divnorm is approximately 50% slower than the baseline, yet the performance gain is 2%. I would like to know whether similar gains can be achieved by training a *compute-matched baseline*. This can be done by either increasing the parameters to match the compute or simply training for more epochs such that overall compute is matched. If this baseline is not included, I understand that the CH-divnorm is significantly more efficient than the Miller divnorm, but it is unclear whether it is still better than just training the baseline for longer.

3. More generally, I find the overall message and contribution to be a bit unclear. The authors mention in a few places that the motivation comes from biology and neuroscience of visual cortex yet the authors still use very implausible networks and do not evaluate the model's ability to predict aspects of cortical responses. On the other hand, these results do not seem strong enough to impact the deep learning field as there are many missing comparisons especially with regards to scale. I would suggest that the authors focus more on the first line of work and try to build a better model of visual cortex with this incorporation of CH-divnorm. That would be a significant contribution to neuroscience that can enable a lot more work. This is especially true when we consider that scale and use of Vision Transformers etc. have improved robustness on ImageNet-C and other benchmarks by much larger margins than this work.




[1] Ren, Mengye, et al. "Normalizing the normalizers: Comparing and extending network normalization schemes." arXiv preprint arXiv:1611.04520 (2016).

[2] Kubilius, Jonas, et al. "Cornet: Modeling the neural mechanisms of core object recognition." BioRxiv (2018): 408385.

**Questions:**

Weaknesses are above. In addition to that I have one minor question:
1. I don't fully understand the interpretation of Figure 2. The authors claim that local opponency doesn't emerge in Figure 2a) but does with the introduction of CH-divnorm. Could you clarify what you mean because I see that the cells become grouped (in the grid differently), but the individual filters that are learned look quite similar across all three subpanels. And many of the filters in Figure 2(a) are color opponent unless I am missing something?

---

### Meta-Review · Area_Chair_cqqT · 2024-12-07

**Metareview:**

The paper proposes a local divisive normalization method for CNNs with the goal to improve performance.

The paper got four reviews, and three recommend to reject the paper. The reviewers point to limited experiments, insufficient comparisons, and an unclear contribution, and identified clarity issues. The authors did not respond to the reviewer's questions and concern.

**Additional Comments On Reviewer Discussion:**

The authors did not respond to the reviewer's questions and concern.

---

### Decision · Program_Chairs · 2025-01-22

Reject